# Teleportation With Null Space Gradient Projection for Optimization Acceleration

## Abstract

Optimization techniques have become increasingly critical due to the ever-growing model complexity and data scale. In particular, teleportation has emerged as a promising approach, which accelerates convergence of gradient descent-based methods by navigating within the loss invariant level set to identify parameters with advantageous geometric properties. Existing teleportation algorithms have primarily demonstrated their effectiveness in optimizing Multi-Layer Perceptrons (MLPs), but their extension to more advanced architectures, such as Convolutional Neural Networks (CNNs) and Transformers, remains challenging. Moreover, they often impose significant computational demands, limiting their applicability to complex architectures. To this end, we introduce an algorithm that projects the gradient of the teleportation objective function onto the input null space, effectively preserving the teleportation within the loss invariant level set and reducing computational cost. Our approach is readily generalizable from MLPs to CNNs, transformers, and potentially other advanced architectures. We validate the effectiveness of our algorithm across various benchmark datasets and optimizers, demonstrating its broad applicability.

## 1 Introduction

Consider an optimization problem where the objective function, denoted by $\mathcal{L}(\omega)$, is parameterized by $\omega \in \Omega$. When $\mathcal{L}(\omega)$ is non-convex, gradient-based methods are commonly used to find a set of parameters corresponding to local minimums in the loss landscape. The standard update rule for gradient descent is given by:

$$\omega_{t+1} \leftarrow \omega_t - \eta \nabla \mathcal{L}(\omega_t), \tag{1}$$

where $\omega_t$ represents the parameter values at ieration $t$ and $\eta > 0$ is the learning rate. As a first-order method, gradient descent is computationally efficient but often suffers from slow convergence. In contrast, second-order methods, such as Newton's method, incorporate higher-order geometric information, resulting in faster convergence. However, this comes with significant computational cost, particularly due to the need to compute and invert the Hessian matrix (Hazan, 2019). To this end, **teleportation** is motivated by the need to leverage higher-order geometry while relying only on gradient information.

Teleportation is based on the premise that multiple points in the parameter space can yield the same loss, which forms the **loss invariant level set** of parameters (Du et al., 2018; Kunin et al., 2020). This assumption is particularly feasible in modern deep learning, where most advanced models are highly over-parameterized (Sagun et al., 2017; Tarmoun et al., 2021; Simsek et al., 2021). By identifying the level set, parameters can be teleported within it to **enhance the gradient norm**, thereby accelerating the optimization process (Kunin et al., 2020; Grigsby et al., 2022).

**Related Work.** Zhao et al. (2022) indicates that the behavior of teleportation, despite utilizing only gradient information, closely resembles that of Newton's method. An alternative perspective on teleportation is that it mitigates the locality constraints of the gradient descent algorithm, resembling the dynamics of *warm restart algorithms* (Loshchilov & Hutter, 2016; Dodge et al., 2020; Bouthillier et al., 2021; Ramasinghe et al., 2022). Under this context, each step of gradient descent is equivalent to a proximal mapping (Combettes & Pesquet, 2011). Teleportation periodically relaxes the proximal restriction, allowing the algorithm to restart at a distant location with desirable geometric

properties. *Compared to warm restart algorithms, teleportation incurs minimal to no increase in loss while providing greater control over the movement of parameters.* Notably, the field of teleportation reveals a gap between theoretical developments and practical applications. Zhao et al. (2022) shows that gradient descent (GD) with teleportation can achieve mixed linear and quadratic convergence rates on strongly convex functions. Mishkin et al. (2024) proves that, for convex functions with Hessian stability, GD with teleportation attains a convergence rate faster than $O(1/K)$. ***However, both approaches encounter limitations when applied to empirical studies involving highly non-convex functions, which are a common characteristic of modern architectures***. Specifically, Zhao et al. (2022) develops a symmetry teleportation algorithm *only for Multi-Layer Perceptrons* (MLPs) using group actions (Armenta & Jodoin, 2021; Ganev et al., 2021; Armenta et al., 2023) . However, challenges persist in terms of its generalizability to other contemporary architectures and its relatively low efficiency. Mishkin et al. (2024), on the other hand, tackled a sequential quadratic programming by using linear approximations of the level set, which can *lead to error accumulation* when the architecture becomes more complicated and the number of teleportation steps increase (see Figure 1 for a visual comparision). Moreover, both studies have primarily concentrated on empirical results involving MLPs and the vanilla Stochastic Gradient Descent (SGD) optimizer.

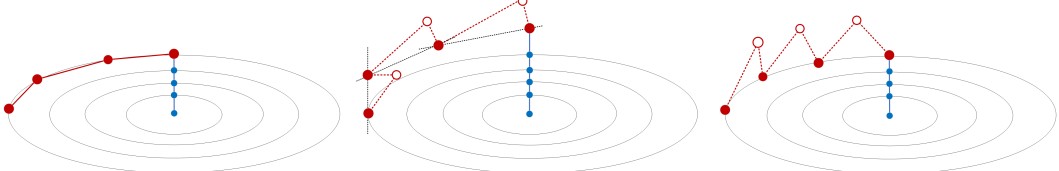

Figure 1: From left to right: symmetry teleport (slow and limited to MLPs), linear approximation of level set (prone to error), our algorithm that projects gradient onto the input null space (fast and accurate).

**Contributions.** Our work seeks to overcome these challenges by designing an algorithm not only ***generalizes to other modern architectures***, but also is ***efficient and accurate***. To be more specific, we eliminate the need for the bottleneck group action transformations of Zhao et al. (2022) by utilizing a more efficient ***gradient projection*** technique. Moreover, instead of taking on the errors introduced by linear approximations of the level set, we ***project the gradient of the teleportation objective onto the input null space***, ensuring an accurate search on the level set thus minimal to no change in loss value. Specifically, our contributions are:

- We propose a novel algorithm that utilizes gradient projection to offer improved computational efficiency and parallelization capabilities.

- The proposed algorithm is a ***general framework that can be easily applied to various modern architectures***, including MLPs, Convolutional Neural Networks (CNNs), transformers, and potentially linear time series models such as Mamba (Gu & Dao, 2023) and TTT (Sun et al., 2024). As a result, our work is the first work to extend teleportation to CNNs and transformers.

- We present ***extensive empirical results*** to demonstrate its effectiveness, spanning a range of benchmark datasets, including MNIST, FashionMNIST, CIFAR-10, CIFAR-100, Tiny-ImageNet, multi-variate time series datasets (electricity and traffic), and Penn Treebank language dataset. We also evaluate the algorithm with multiple modern optimizers, such as SGD (Robbins & Monro, 1951), Momentum (Polyak, 1964), Adagrad (Duchi et al., 2011), and Adam (Kingma, 2014), whereas previous studies primarily focused on the vanilla SGD.

## 2 PRELIMINARY

### 2.1 SYMMETRY TELEPORTATION

In this section, we describe the general framework of teleportation through a state-of-the-art algorithm, *symmetry teleportation* (Zhao et al., 2022; 2023).

Let $G$ be a set of symmetries that preserves the loss value $\mathcal{L}$, i.e., let $\omega = (X, W)$,

$$\mathcal{L}(X, W) = \mathcal{L}(g \cdot (X, W)), \forall g \in G, \tag{2}$$

where $X$ represents data and $W$ represents *parameters of the deep learning model*. Define a teleport schedule $K \subset \{0, 1, ..., T_{max}\}$, where $T_{max}$ is the maximum training epochs. Prior to each epoch in $K$, teleportation is applied by searching for $g \in G$ which transforms the parameter $W$ to $W^*$ with greater gradient norm *within the loss invariant level set*.

When the group $G$ is continuous, the search process can be conducted by parameterizing the group action $g$ and performing gradient ascent on $g$ with the teleportation objective function defined as the gradient norm of the current parameter $W$. For example, general linear group transformations $g \in GL_d(\mathbb{R})$ can be parameterized as $g = I + \epsilon M$, where $\epsilon \ll 1$ and $M$ is an arbitrary matrix.

Zhao et al. (2022; 2023) designs a loss invariant group action *specifically for MLPs with bijective activation function* $\sigma$. Assuming the invertibility of $(k-2)$-th layer's output, $h_{k-2}$, the following group action $g \in GL_d(\mathbb{R})$ on $k$-th and $(k-1)$-th layers ensures the output of the entire network unchanged:

$$g_m \cdot W_k = \begin{cases} W_m g_m^{-1} & \text{if } k = m, \\ \sigma^{-1} \left(g_m \sigma \left(W_{m-1} h_{m-2}\right)\right) h_{m-2}^{-1} & \text{if } k = m-1, \\ W_k & \text{if } k \notin \{m, m-1\}. \end{cases}$$

In practice, each teleportation update applies the above group action to every layer of an MLP, requiring two bottleneck inverse operations per update. Denote $D_{max}$ as the largest width of the MLPs, and $n$ the sample size, assuming $D_{max} > n$. The time complexity of calculating pseudo-inverse for each layer is $O(D_{max}^2 n)$. Therefore, the total time complexity for $l$ layers, $b$ batches, and $t$ teleport updates per batch is $O(D_{max}^2 n l b t)$. The need for pseudo-inverse computations and the dependencies between layers render the algorithm relatively slow and unsuitable for parallelization. Additionally, there is no straightforward method to generalize this design from MLPs to CNNs or transformers.

## 2.2 Matrix Approximation With SVD

An arbitrary matrix $A \in \mathbb{R}^{(m,n)}$ can be decomposed using the singular value decomposition (SVD) Klema & Laub (1980) as $A = U\Sigma V^T$, where $U \in \mathbb{R}^{(m,m)}$ consists of orthonormal eigenvectors of $AA^T$, $\Sigma \in \mathbb{R}^{(m,n)}$ is a diagonal matrix containing sorted singular values, and $V \in \mathbb{R}^{(n,n)}$ contains orthonormal eigenvectors of $A^T A$. The matrix $A$ can be expressed as $\sum_{i=1}^{r} \sigma_i u_i v_i^T$, where $r = \min(m, n)$, and $(u_i, v_i)$ are the column and row vectors of $U, V$ respectively.

In this work, we consider the matrix approximation $A_k$ of $A$ defined as $A_k = \sum_{i=1}^{k} \sigma_i u_i v_i^T$, where

$$k = \arg\min_k \left\{ k : ||A_k||_F^2 \geq \tau ||A||_F^2 \right\}, \tag{3}$$

with $\|\cdot\|_F$ denotes the Frobenius norm and $\tau \in [0, 1]$ being a threshold hyper-parameter.

## 3 Teleport With Null Space Gradient Projection

Our objective is to develop a generalizable and efficient algorithm that avoids reliance on specific group action designs. Moreover, it should avoid any (linear) approximation of the level set with uncontrollable errors, as these could otherwise result in suboptimal performance. Considering the common architectural design in modern neural networks, which typically employ a linear relationship between weights and inputs of each layer, the technique of ***gradient projection on to the input null space*** of each layer is well-suited for this purpose. We next elaborate on it.

**Gradient Projection.** To incorporate the geometric landscape and accelerate optimization using only gradient information, the objective function for teleportation is defined as the squared gradient norm of the loss function of the primary task with respect to the model parameter $W$,

$$L_{teleport} = \frac{1}{2} \|\nabla_W L_{primary}\|^2. \tag{4}$$

During each teleportation step, in contrast to symmetry teleportation, the gradient ascent is applied directly on the model parameter $W_l$ of each layer $l$ instead of relying on an intermediate group action $g$, i.e., we have

$$W_{l,t+1} = W_{l,t} + \eta \pi_l(\nabla_{W_l} L_{teleport}),  \tag{5}$$

where $\eta$ is the learning rate for teleportation update, and $\pi_l$ is the **layerwise projection operator** onto the null space of each layer's input. We have distinct projection operators for different model architectures. ***We will derive $\pi_l$ for MLPs, CNNs and transformes in the sequel***. The validity of this projection is based on the assumption that *the gradient resides within the span of each layer's input for certain structures*, which will also be elaborated in a subsequent section.

**Section Organization.** We first define and provide notations for MLPs, CNNs, and transformers. Next, we demonstrate that the gradient in Equation 5 indeed resides within the input space of these architectures, thus ***satisfying the required assumption of gradient projection***. Finally, we present our proposed approach and provide a detailed explanation of how to derive the projection operators for each of these architectures.

### 3.1 DEEP LEARNING ARCHITECHTURES

#### 3.1.1 MULTI-LAYER PERCEPTRONS

We define the $l$-th layer of an MLP (Rumelhart et al., 1986). Denote the input of the layer as $x_{l-1} \in \mathbb{R}^{(d_{l-1}, 1)}$, the parameter as $W_l \in \mathbb{R}^{(d_l, d_{l-1})}$, the output as $x_l \in \mathbb{R}^{(d_l, 1)}$. We incorporate the bias term into $W_l$ and $x_{l-1}$ by adding an additional column to $W_l$ and unity to $x_{l-1}$. Then the output of $l$-th layer is defined as

$$x_l = \sigma(W_l x_{l-1}),$$

where $\sigma$ is an activation function, e.g. ReLU (Nair & Hinton, 2010).

#### 3.1.2 CONVOLUTIONAL NEURAL NETWORK

We define the $l$-th layer of a CNN (LeCun et al., 1998). Denote the input to the $l$-th convolutional layer as $x_{l-1} \in \mathbb{R}^{C_i \times h_i \times w_i}$, convolutional kernel as $W_l \in \mathbb{R}^{C_o \times C_i \times k \times k}$, and output as $x_l \in \mathbb{R}^{C_o \times h_o \times w_o}$, where $C_i, h_i, w_i (C_o, h_o, w_o)$ are the input (output) channel, height, and width, respectively, and $k$ is the kernel size. If $x_{l-1}$ (e.g., with padding, striding, etc) is reshaped into $(h_o \times w_o) \times (C_i \times k \times k)$ as $X_{l-1}$, and $W_l$ is reshaped to $(C_i \times k \times k) \times C_o$, then the convolutional layer can be expressed as a matrix multiplication

$$x_l = \sigma(X_{l-1} W_l),$$

where $x_l \in \mathbb{R}^{(x_o \times w_o) \times C_o}$ is the output of $l$-th layer, and $\sigma$ an activation function. See Appendix A.3 for a visual explanation of the matrix multiplication.

#### 3.1.3 TRANSFORMER

We define the self-attention and multi-head self-attention layers (Vaswani, 2017). Denote the input sequence of the $l$-th self-attention layer as $X_{l-1} \in \mathbb{R}^{T \times D_i}$, with sequence length $T$ and dimension $D_i$. The $l$-th self-attention layer is parameterized by the query matrix $W_{l,q} \in \mathbb{R}^{(D_i, D_k)}$, the key matrix $W_{l,k} \in \mathbb{R}^{(D_i, D_k)}$, and the value matrix $W_{l,v} \in \mathbb{R}^{(D_i, D_o)}$. Then, the self-attention layer maps the sequence from dimension $D_i$ to $D_o$ by

$$Attention(Q, K, V) = softmax(\frac{QK^T}{\sqrt{D_k}})V,$$

where $Q = X_{l-1} W_{l,q}, K = X_{l-1} W_{l,k}, V = X_{l-1} W_{l,v}$, and $D_k$ is the dimension of the model.

The multi-head attention is realized by replicating and concatenating $N_h$ heads of low-rank self-attentions before applying an output projection, defined as

$$MultiHead(X_{l-1}) = concat_{i \in [N_h]}[H^{(i)}]W_{l,o}  \tag{6}$$

$$H^{(i)} = Attention(X_{l-1} W_{l,q}^{(i)}, X_{l-1} W_{l,k}^{(i)}, X_{l-1} W_{l,v}^{(i)}),  \tag{7}$$

where $W_{l,q}^{(i)} \in \mathbb{R}^{(D_i, \frac{D_k}{N_h})}$, $W_{l,k}^{(i)} \in \mathbb{R}^{(D_i, \frac{D_k}{N_h})}$, $W_{l,v}^{(i)} \in \mathbb{R}^{(D_i, \frac{D_k}{N_h})}$ are parameters for each head. The output projection matrix $W_{l,o} \in \mathbb{R}^{(D_k, D_o)}$ maps the concatenation of heads to the desired output dimension.

## 3.2 INPUT AND GRADIENT SPACE

Now we establish that ***the gradient of the teleportation objective function resides within the space spanned by the input of each layer***. Following the notation established in Section 3.1, we can readily express the gradient of the teleportation objective function with respect to the model parameter $W_l$:

$$\text{MLP} : \nabla_{W_l} L_{Teleport} = \nabla_{(W_l x_{l-1})} L_{Teleport} \cdot \nabla_{W_l}(W_l x_{l-1})$$
$$= \delta_{MLP} x_{l-1}^T$$
$$\text{CNN} : \nabla_{W_l} L_{Teleport} = \nabla_{W_l}(X_{l-1} W_l) \cdot \nabla_{(X_{l-1} W_l)} L_{Teleport}$$
$$= X_{l-1}^T \cdot \delta_{CNN}$$
$$\text{Self-Attention} : \nabla_{W_{l,\cdot}^{(i)}} L_{Teleport} = \nabla_{W_{l,\cdot}^{(i)}}(X_{l-1} W_{l,\cdot}^{(i)}) \cdot \nabla_{(X_{l-1} W_{l,\cdot}^{(i)})} L_{Teleport}$$
$$= X_{l-1}^T \cdot \delta_{Attention},$$

where $\delta_{MLP} \in \mathbb{R}^{(d_l, 1)}$, $\delta_{CNN} \in \mathbb{R}^{(h_o \times w_o, C_o)}$, and $\delta_{Attention} \in \mathbb{R}^{(T, D_k)}$ are some error terms determined by both the loss function of the primary task and the objective function of the teleportation. Here, it can be observed that all gradients above can be written as the matrix multiples involving the input $X$ of each layer and another matrix. Thus, the gradient of the teleportation objective function indeed resides within the space spanned by the input of each layer for MLPs, CNNs, and transformer, which is a composition of attention layers and MLP layers.

## 3.3 ALGORITHM

**Step 1.** We first construct the representation matrix for each layer $l$ based on a given teleportation batch of data:

$$\text{MLP} : R_{MLP}^l = [x_{l-1,1}, x_{l-1,2}, \cdots, x_{l-1,n}] \tag{8}$$
$$\text{CNN} : R_{CNN}^l = [X_{l-1,1}^T, X_{l-1,2}^T, \cdots, X_{l-1,n}^T] \tag{9}$$
$$\text{Self-Attention} : R_{Attention}^l = [X_{l-1,1}^T, X_{l-1,2}^T, \cdots, X_{l-1,n}^T], \tag{10}$$

where $n$ is the batch size. Each representation matrix $R_{MLP}^l \in \mathbb{R}^{(d_{l-1}, n)}$, $R_{CNN}^l \in \mathbb{R}^{(C_i \times k \times k, h_o \times w_o \times n)}$, and $R_{Attention}^l \in \mathbb{R}^{(D_i, T \times n)}$ contains columns of feature vectors, which are captured at each layer during the forward pass through the network using a random teleportation batch of size $n$.

**Step 2.** For all model architectures, we apply SVD on the representation matrix $R^l$, followed by a low-rank approximation $(R^l)_k = \sum_{i=1}^k \sigma_{l,i} u_{l,i} v_{l,i}^T$ based on the criterion in Equation 3, using a predefined threshold $\tau$. The orthonormal column vectors $[u_{l,1}, u_{l,2}, \ldots, u_{l,k}]$, from SVD of $R^l$, consist of the eigenvectors corresponding to the top $k$ singular values of the representation matrix. We define the subspace spanned by these eigenvectors as ***the space of significant representation*** (Saha et al., 2021b).

During a teleportation step, the goal is to ensure that the gradient update in Equation 5 preserves the correlation between the weights and the space of significant representation as much as possible. Given that the gradient space lies within the input space, we can partition the gradient space into two orthogonal subspaces of the input space: the ***Core Gradient Space (CGS)*** and the ***Residual Gradient Space (RGS)*** (Saha et al., 2021a), which are spanned by $[u_{l,1}, u_{l,2}, \cdots, u_{l,k}]$ and $[u_{l,k+1}, u_{l,k+2}, \cdots, u_{l,r}]$ respectively. By construction, projecting the gradient onto CGS will lead to the greatest interference in the correlation between the weights and the space of significant representation, while ***projecting onto RGS will result in minimal or no interference in this correlation***. To preserve model parameters on the loss-invariant level set during teleportation steps, we project the gradient of teleportation objective function $\nabla_W L_{Teleport}$ onto the RGS before each update.

**Step 3.** Given the orthonormal basis $B_l = [u_{l,1}, u_{l,2}, \cdots, u_{l,k}]$ of the CGS for the $l$-th layer, the gradient $\nabla_{W_l} L_{Teleport}$ is initially projected onto the CGS and then removed from itself to yield the projection onto the RGS. Specifically, the projection operator $\pi_l$ is defined as follows:

$$\text{MLP} : \pi_l(\nabla_{W_l} L_{Teleport}) = \nabla_{W_l} L_{Teleport} - (\nabla_{W_l} L_{Teleport}) B_l B_l^T \tag{11}$$

$$\text{CNN} : \pi_l(\nabla_{W_l} L_{Teleport}) = \nabla_{W_l} L_{Teleport} - B_l B_l^T (\nabla_{W_l} L_{Teleport}) \tag{12}$$

$$\text{Self-Attention} : \pi_l(\nabla_{W_{l,\cdot}^{(i)}} L_{Teleport}) = \nabla_{W_{l,\cdot}^{(i)}} L_{Teleport} - B_l B_l^T (\nabla_{W_{l,\cdot}^{(i)}} L_{Teleport}) \tag{13}$$

The teleportation step is completed by substituting the projection operator back into Equation 5. The complete training process is outlined in the pseudo-code presented in appendix A.1.

## 4 EXPERIMENTS

In this section, we demonstrate the effectiveness of our method across MLPs, CNNs, and transformers, utilizing *a wide range of benchmark datasets*. Additionally, we evaluate our approach using *a variety of optimizers*, such as the vanilla SGD, first-moment optimizer like SGD with momentum, second-moment optimizers like Adagrad and Adam.

We showcase the efficiency of our algorithm compared to the state-of-the-art method, symmetry teleportation, across multiple teleportation hyperparameters. Furthermore, if any approximation of the level set is needed, we demonstrate the *capability of our approach to control the error in null space approximation*, which subsequently improves the accuracy of level set approximation during the teleportation.

### 4.1 MLP EXPERIMENTS

**Datasets.** To demonstrate the effectiveness of our method with MLPs, we conduct experiments using the MNIST digit image classification dataset and its clothing variant, FashionMNIST. Both datasets are split into $60,000$ samples for training and $10,000$ samples for testing. The input images, with dimensions of $28 \times 28$ pixels, are flattened into vectors before being fed into the MLPs models.

**Implementation Detail.** We use a 3-layer MLPs with hidden dimensions [1024, 1024], ReLU activation function, and cross-entropy loss. Following the convention in Zhao et al. (2022)'s work, we schedule teleportation for the first 5 epochs of the primary training phase. For each teleportation in the schedule, we randomly sample 32 batches of data and perform 8 teleport updates per batch. The SVD threshold is set to 1, i.e., *the gradients are projected onto the exact input null space*. Learning rates are set differently depending on the optimizer used. See the appendix A.2 for complete implementation details.

**Experiment Results.** With teleportation, in Figure 2, we observe a faster convergence rate for both training and test loss, ultimately converging to a lower loss compared to their non-teleportation counterparts. This behavior suggests that teleportation may have the potential to not only accelerate the convergence rate but also help in finding a better local minimum.

### 4.2 CNN EXPERIMENTS

**Datasets.** We use the CIFAR-10, CIFAR-100, and Tiny-Imagenet datasets to evaluate the effectiveness of our algorithm on CNNs. Both CIFAR datasets are split into 50,000 training samples and 10,000 test samples. The image size for CIFAR datasets is $3 \times 32 \times 32$. The Tiny-Imagenet dataset is a smaller version of the full Imagenet dataset, containing 200 image classes with $100,000$ training images and $20,000$ validation/test images. The image size for the Tiny-Imagenet dataset is kept the same as the full Imagenet dataset, i.e., $3 \times 224 \times 224$.

**Implementation Detail.** For the CIFAR datasets, we use a 3-layer CNNs with channels $[3, 16, 32, 64]$, max pooling after each layer, ReLU activation function, and cross-entropy loss. For the Tiny-Imagenet dataset, we utilize a residual network with channels $[3, 64, 64, 64, 128, 128, 128, 256, 256, 256]$, and 3 residual connections between channels of same shape. Instead of max pooling, we use larger strides to reduce the feature size, a common practice in the design of residual networks. A classification head is connected after the final channel for

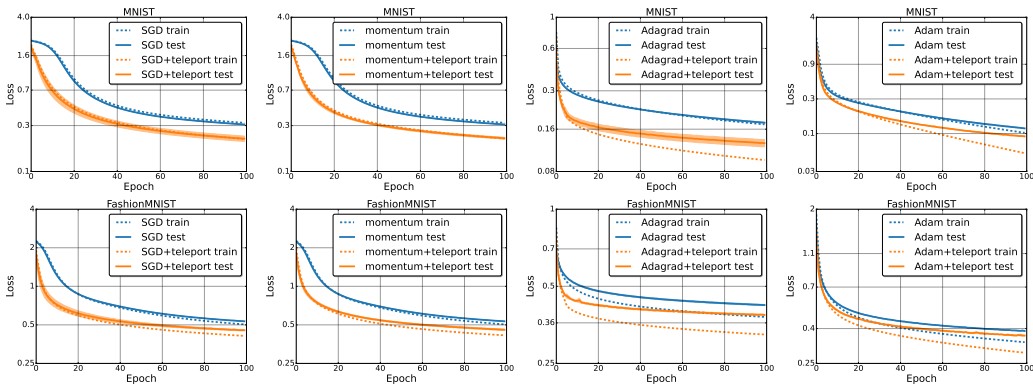

Figure 2: Loss trajectories of training MLPs on the MNIST and FashionMNIST datasets. Each experiment is repeated 3 times, with the average loss plotted and the standard deviation of loss represented as the shaded area.

both architectures. The teleportation scheduling and threshold $\tau$ remains the same as in the MLPs experiments. See appendix A.2 for complete implementation details.

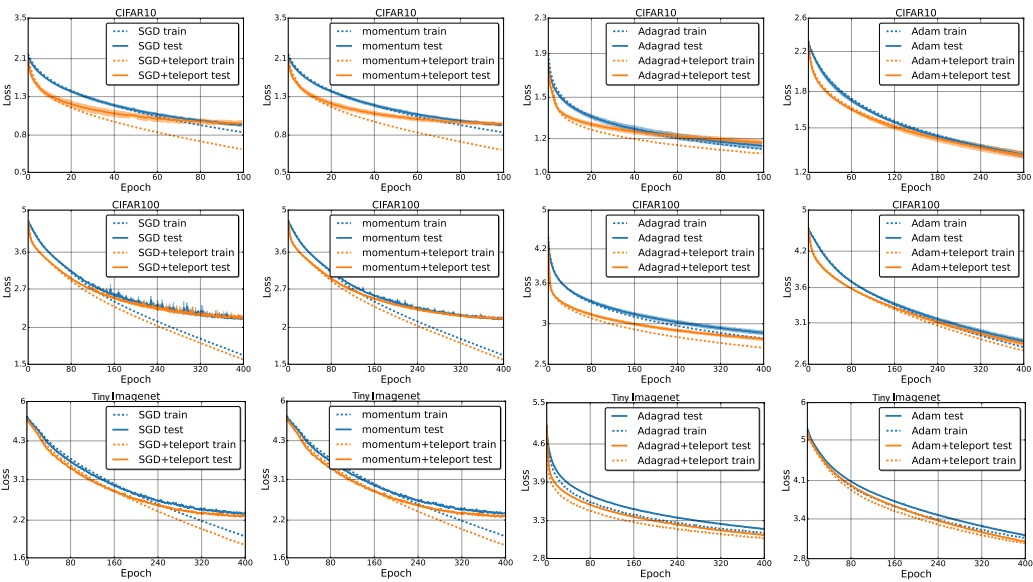

Figure 3: Loss trajectories of training CNNs on CIFAR datasets and Tiny-Imagenet dataset. Each experiment is repeated 3 times, with the average loss plotted and the standard deviation of loss represented as the shaded area.

**Experiment Results.** With teleportation, we observe in Figure 3 a marked acceleration in optimization in the beginning of each training, coinciding with the application of teleportation. The test loss with teleportation tends to converge to the same value as the non-teleportation counterpart, while the training loss with teleportation continues to decrease at a faster rate even after the test loss has plateaued. This behavior is expected, as the teleportation objective is defined as the squared norm of the gradient, which prioritizes faster convergence on the training set rather than improving generalization. The teleportation framework is highly flexible, allowing the teleportation objective function to be adjusted to other reasonable choices, such as the curvature of the parameter landscape, which has been shown to enhance generalization (Zhao et al., 2023).

### 4.3 TRANSFORMER EXPERIMENTS

**Datasets.** We first consider the MNIST dataset as a sequential classification task, with a sequence length of $28 \times 28$ and a data dimension $1$.

Next, we evaluate on two publicly available multi-variate time series regression datasets: electricity and traffic. The electricity dataset consists of $321$ dimensions with a total sequence length of $26,304$. The sample sequence length is set to $7 \times 24$, representing a week's worth of data. The regression target is the data point of the same dimension $24$ hours after the input sample. The traffic dataset consists of $862$ dimensions, with a total sequence length of $17,544$. The data is similarly manipulated to regress a week's worth of data to the data $24$ hours after the week. See Appendix A.4 for a detailed explanation.

We also evaluate on the Penn Treebank (PTB) language corpus. We use the default train/test split of the PTB dataset, where the training set contains approximately $950,000$ words and the test set approximately $80,000$ words. We use the TreebankWord tokenizer from the nltk Library and set the sequence length to $256$. As is common practice, we formulate the problem as a causal self-supervised learning task, where the label is the input shifted to the right by one.

**Implementation Detail.** For the sequential MNIST dataset, we use a small Transformer model with 2 heads, each having a dimension of $64$, stacked across two layers. For the regression and language datasets, we use a transformer with 4 heads, each with a dimension of $64$, stacked across $4$ layers without pooling, followed by a linear output. See appendix A.2 for complete implementation details.

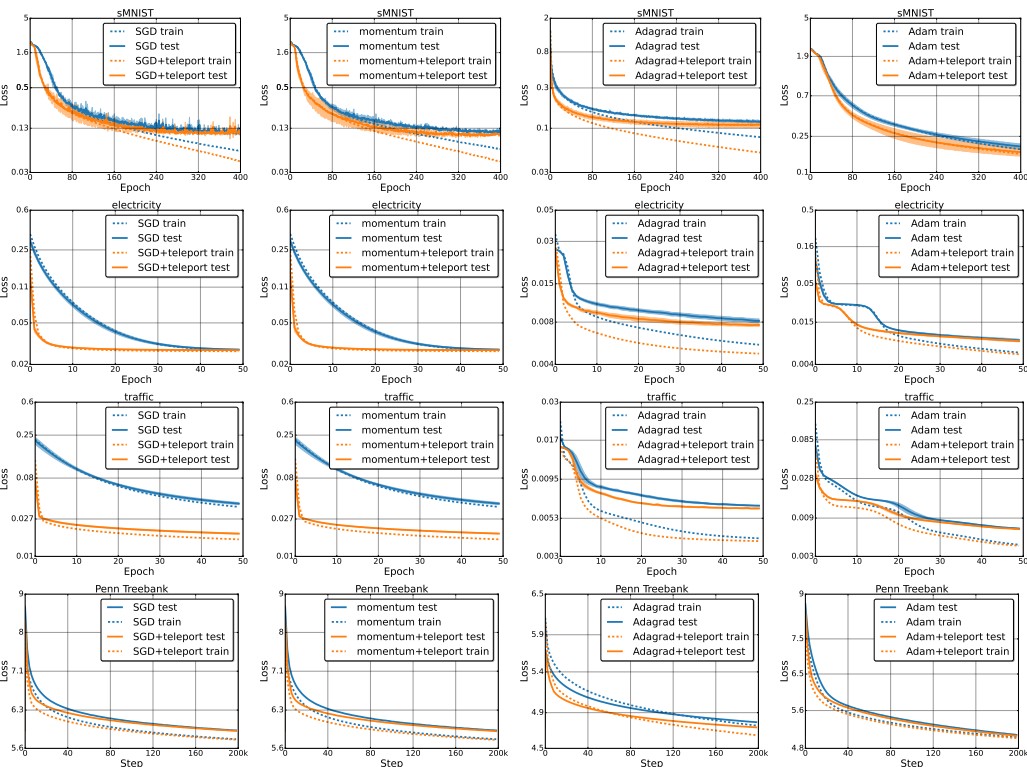

Figure 4: Loss trajectories of training Transformers on sequential MNIST, electricity, traffic, and Penn Treebank datasets. Each experiment is repeated 3 times, with the average loss plotted and the standard deviation of loss represented as the shaded area.

**Experiment Results.** In addition to the observations from previous experiments, in Figure 4, we notice that teleportation remains effective across different problem settings, including regression problems and language modeling. Significant acceleration is observed in the regression datasets,

particularly with the SGD and momentum optimizers, where the loss with teleportation converges within the first few epochs, while the non-teleportation counterpart takes more than 50 epochs to converge on the traffic dataset. Furthermore, the acceleration with teleportation in language modeling is particularly notable during the initial phase of training, even though both approaches eventually converge to the same loss. These results highlight the potential of applying teleportation to the training of large language models.

## 4.4 EFFICIENCY IMPROVEMENT

In this section, we demonstrate the efficiency of our algorithm compared to the state-of-the-art symmetry teleportation algorithm.

Recall that the time complexity of symmetry teleportation is $O(d^2nlbt)$, where $d$ is the feature dimension of layers, $n$ is the batch size, $l$ is the number of layers, $b$ is the number of batches, and $t$ is the number of teleport steps per batch. Note that the pseudo-inverse is calculated using SVD for Pytorch Library, thus sharing the same time complexity as SVD operation. However, in our method, only one SVD is needed for each batch of data, which reduces the bottleneck and brings the time complexity down to $O(d^2nlb)$. Ideally, by leveraging our algorithm's layer-independent property, computations can be parallelized across all layers, further reducing the time complexity to $O(d^2nb)$. However, we leave such engineering optimizations for future work.

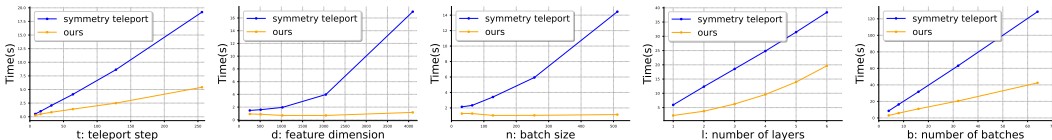

Figure 5: From left to right: a comparison between symmetry teleport and our algorithm using MLPs in terms of the scaling of runtime with respect to $t$, $d$, $n$, $l$, and $b$.

In practice, as demonstrated in Figure 5, our algorithm exhibits linear scaling with respect to $t$, $l$, and $b$, while the runtime of the symmetry teleportation increases at a significantly faster rate. Notably, for $d$ and $n$, our approach achieves near-constant runtime in contrast to the linear-to-polynomial runtime of the symmetry teleport. Ideally, once the layer parallelization is fully implemented, we anticipate that constant runtime will also be achieved with an increasing number of layers, thereby enhancing overall performance.

## 4.5 ERROR CONTROL

In addition to its efficiency, our algorithm provides a distinct advantage in controlling the error associated with increased loss during teleportation. Figure 6a records the information of the input space of the second layer in MLPs, CNNs, and Transformers (with the same architechtures used in experiments) across all datasets. Most variance of input is captured by the space of significant representation of a relatively small proportion of total dimensions, represented by the percentages of sorted eigenvectors in SVD. Consequently, even without approximating the input null space, sufficient dimensions are typically available in the null space to facilitate gradient projection and search. ***This validates our choice of setting $\tau$ to be $1$ in most cases.*** Figure 6b further confirms that when the threshold $\tau$ is set to 1, meaning the exact null space is utilized, the gradient norm increases steadily during teleportation while the loss remains constant. Moreover, as $\tau$ decreases, the gradient is projected onto an approximated null space with a significantly larger number of dimensions, yet capturing only slightly more variance with minimal impact on the loss. A remarkable increase in the gradient norm ascending speed is observed when $\tau$ is set to 0.99, with the loss still remaining constant. (Experiments in Figure 6b are conducted using transformer on sMNIST dataset.)

## 5 DISCUSSION AND CONCLUSION

In this paper, we propose a novel algorithm that generalizes the application of teleportation from MLPs to other modern architectures such as CNNs and transformers. The algorithm demonstrates

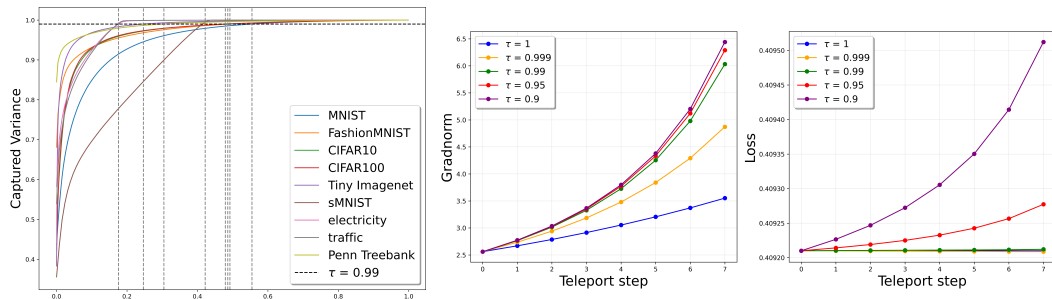

(a) Input variance captured by eigenvectors.

(b) Effect of teleport step on increase of gradient norm and loss value.

Figure 6: A majority of the input variance is captured by a relatively small proportion of the input space. As we approximate a larger input null space, the gradient norm increases more rapidly during teleportation, while the loss remains constant when $\tau$ is greater than $0.99$.

improved computational efficiency and introduces explicit error control during the level set approximation, if such an approximation is employed.

Gradient projection proves to be a powerful tool for modern AI, as most contemporary architectures rely on a linear modeling between inputs and weights. Consequently, our framework has the potential to be generalized to emerging time-series architectures such as Mamba and TTT.

Despite its promising performance, teleportation still faces challenges when applied broadly in the deep learning field. One of the major challenges is the selection of hyperparameters. Identifying a generalizable set of hyperparameters suitable for all architectures and datasets remains difficult. Developing a simple and effective hyperparameter selection strategy will significantly enhance the overall efficiency of teleportation.

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

## A    APPENDIX

### A.1    PSEUDOCODE

---

**Algorithm 1** Teleportation with Input Null Space Gradient Projection

---

**Input:** Loss function $\mathcal{L}(w)$, number of epochs for primary task $T$, teleport learning rate $\eta$, teleport batch number $b$, teleport step number $t$, teleport schedule $K$, threshold maximum gradient norm value CAP, initialized parameters $w_0$.
**Output:** $w_T$.
1: **for** $i \leftarrow 0$ to $T - 1$ **do**
2:     **if** $i \in K$ **then**
3:         **for** $b$ batches **do**
4:             Null space projection matrix $\pi \leftarrow$ SVD(batch)
5:             **for** t steps **do**
6:                 **if** $\|\nabla_w \mathcal{L}|_{w_i}\|^2 <$ CAP **then**
7:                     $w_i \leftarrow w_i - \eta \pi (\nabla_w \|\nabla_w \mathcal{L}|_{w_i}\|^2|_{w_i})$
8:                 **else**
9:                     **break**
10:                **end if**
11:            **end for**
12:        **end for**
13:    **end if**
14:    Continue the training of the primary task
15: **end for**
16: **return** $w_T$

---

### A.2    IMPLEMENTATION DETAILS

In table 1, we summarize the hyper-parameters used in experiments. We denote the base learning rate for primary task as $\eta_{prim}$, the learning rate for teleportation as $\eta_{tele}$, maximum epoch for primary task as $T_{prim}$, teleport batch size as $n$, and teleport cap threshold as CAP. The batch size for the primary task is set to 32, the number of teleport batches set to 32, and the number of teleportation steps per batch set to 8 throughout all experiments.

For all experiments using CNNs, we perform 40 warm-up steps before the first teleportation to stabilize the behavior of the gradients.

For the sequential MNIST dataset, we use a small Transformer model with 2 heads, each having a dimension of 64, stacked across two layers. This is followed by an average pooling layer and a ten-way linear classification head, optimized using cross-entropy loss. For the electricity and traffic datasets, we use a transformer with 4 heads, each with a dimension of 64, stacked across 4 layers without pooling, followed by a linear regression head where the output dimension matches the input dimension. For the PTB dataset, we use the same Transformer architecture but replace the first linear layer with an embedding layer and set the output dimension to the vocabulary size, which is approximately $10,000$.

### A.3    VISUALIZATION OF MATRIX MULTIPLICATION REPRESENTATION FOR CNNS

Although filters in CNNs works differently than weights in MLPs, the forward and backward propagations of CNNs are essentially still matrix multiplications (see Figure 7).

| Dataset (Optimizer) | $\eta_{prim}$ | $\eta_{tele}$ | $T_{prim}$ | n | CAP |
|---|---|---|---|---|---|
| MNIST (SGD) | $2e-4$ | $2e-1$ | 100 | 32 | 5 |
| MNIST (Momentum) | $2e-4$ | $2e-1$ | 100 | 32 | 5 |
| MNIST (Adagrad) | $2e-4$ | $2e-1$ | 100 | 32 | 5 |
| MNIST (Adam) | $2e-4$ | $2e-1$ | 100 | 32 | 5 |
| FashionMNIST (SGD) | $2e-4$ | $2e-1$ | 100 | 32 | 5 |
| FashionMNIST (Momentum) | $2e-4$ | $2e-1$ | 100 | 32 | 5 |
| FashionMNIST (Adagrad) | $2e-4$ | $2e-1$ | 100 | 32 | 5 |
| FashionMNIST (Adam) | $2e-4$ | $2e-1$ | 100 | 32 | 5 |
| CIFAR10 (SGD) | $1e-4$ | $3e-3$ | 100 | 256 | 40 |
| CIFAR10 (Momentum) | $1e-4$ | $3e-3$ | 100 | 256 | 40 |
| CIFAR10 (Adagrad) | $1e-4$ | $3e-3$ | 100 | 256 | 40 |
| CIFAR10 (Adam) | $1e-5$ | $3e-3$ | 300 | 256 | 40 |
| CIFAR100 (SGD) | $1e-4$ | $3e-3$ | 400 | 256 | 40 |
| CIFAR100 (Momentum) | $1e-4$ | $3e-3$ | 400 | 256 | 40 |
| CIFAR100 (Adagrad) | $1e-4$ | $3e-3$ | 400 | 256 | 40 |
| CIFAR100 (Adam) | $3e-5$ | $3e-3$ | 400 | 256 | 40 |
| Tiny Imagenet (SGD) | $2e-4$ | $3e-3$ | 400 | 32 | 40 |
| Tiny Imagenet (Momentum) | $2e-4$ | $3e-3$ | 400 | 32 | 40 |
| Tiny Imagenet (Adagrad) | $2e-4$ | $3e-3$ | 400 | 32 | 40 |
| Tiny Imagenet (Adam) | $5e-5$ | $3e-3$ | 400 | 32 | 40 |
| sMNIST (SGD) | $1e-3$ | $3e-3$ | 400 | 32 | 10 |
| sMNIST (Momentum) | $1e-3$ | $3e-3$ | 400 | 32 | 10 |
| sMNIST (Adagrad) | $1e-3$ | $3e-3$ | 400 | 32 | 10 |
| sMNIST (Adam) | $1e-4$ | $3e-3$ | 400 | 32 | 10 |
| electricity (SGD) | $1e-4$ | $3e-3$ | 50 | 32 | 10 |
| electricity (Momentum) | $1e-4$ | $3e-3$ | 50 | 32 | 10 |
| electricity (Adagrad) | $1e-4$ | $3e-3$ | 50 | 32 | 10 |
| electricity (Adam) | $1e-4$ | $3e-3$ | 50 | 32 | 10 |
| traffic (SGD) | $1e-4$ | $3e-3$ | 50 | 32 | 10 |
| traffic (Momentum) | $1e-4$ | $3e-3$ | 50 | 32 | 10 |
| traffic (Adagrad) | $1e-4$ | $3e-3$ | 50 | 32 | 10 |
| traffic (Adam) | $1e-4$ | $3e-3$ | 50 | 32 | 10 |
| Penn Treebank (SGD) | $2e-4$ | $5e-2$ | 20,000 steps | 32 | 5 |
| Penn Treebank (Momentum) | $2e-4$ | $5e-2$ | 20,000 steps | 32 | 5 |
| Penn Treebank (Adagrad) | $2e-4$ | $5e-2$ | 20,000 steps | 32 | 5 |
| Penn Treebank (Adam) | $5e-5$ | $5e-2$ | 20,000 steps | 32 | 5 |

Table 1: Summary table for hyper-parameters of all experiments

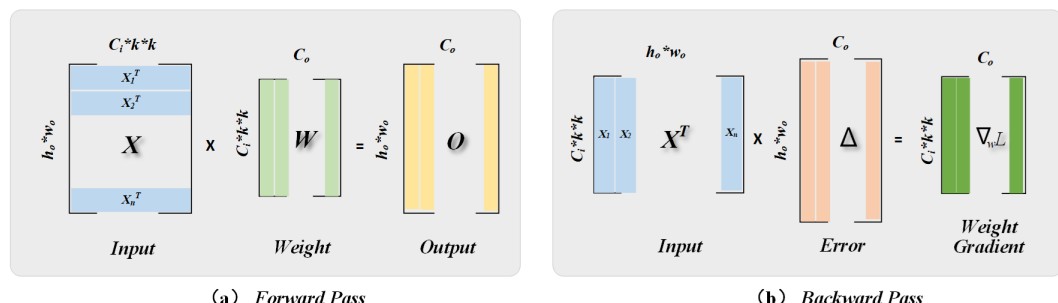

Figure 7: Visualization of matrix representation of forward and backward pass for CNNs.

## A.4 BRIEF EXPLANATION OF THE MULTI-VARIATE TIME SERIES REGRESSION DATASETS

The electricity dataset tracks electricity consumption in kWh every $15$ minutes from $2012$ to $2014$ for $321$ clients, adjusted to reflect hourly consumption. The dataset consists of $321$ dimensions with a total sequence length of $26,304$. The sample sequence length is set to $7 \times 24$, representing a week's worth of data. The regression target is the data point of the same dimension $24$ hours after the input sample. The traffic dataset contains $48$ months ($2015$–$2016$) of hourly data from the California Department of Transportation, describing road occupancy rates (between $0$ and $1$) measured by various sensors on the San Francisco Bay Area freeway. This dataset consists of $862$ dimensions, with a total sequence length of $17,544$. The data is similarly manipulated to regress a week's worth of data to the data $24$ hours after the week.

