# OpenReview forum: "TELEPORTATION WITH NULL SPACE GRADIENT PROJECTION FOR OPTIMIZATION ACCELERATION"
_ICLR.cc/2025/Conference — Submitted to ICLR 2025_

### Official Review · Reviewer_KxS7 · 2024-11-02

**Soundness:** 2
**Presentation:** 1
**Contribution:** 2
**Rating:** 3
**Confidence:** 3

**Summary:**

This paper proposes an improved algorithm for teleportation-based optimization, in which at certain times during training the network weights are moved to a different location with roughly the same loss but higher gradient norm. Their approach uses an approximate projection onto the loss level set while optimizing the gradient norm during the teleportation step. The algorithm is validated on experiments with MLPs, CNNs, and Transformers.

**Strengths:**

1. The authors propose a faster algorithm for the teleportation step that can also be applied to architectures beyond MLPs.
2. The authors validate their approach by using it to modify a variety of standard optimizers on multiple problems, showing consistent improvements in training loss.
3. Code is attached to reproduce the results.

**Weaknesses:**

1. The approximation scheme is not very clearly described. It would be helpful to have pseudocode for the actual core contribution of the work, rather than just for the teleportation approach more broadly. The goal of the approximation (“to ensure that the gradient update in Equation 5 preserves the correlation between the weights and the space of significant representation as much as possible”) is only defined in words, and the error we should expect is not analyzed.
2. A major claim is the improved speed over past teleportation approaches, such as the one using symmetry or the one using linear approximation. However, no comparison is done for either case. For example, it would be useful to see whether the reduced per-iteration complexity is enough to overcome the approximation error and to do better than the symmetry-based approach. Similarly, a wallclock (rather than epoch) comparison between this approach and vanilla optimizers would also be useful.
3. The observed improvement is mainly in training performance, with minimal improvement in generalization. The authors suggest that there may be ways to overcome that, but do not evaluate this. Without a clear path towards test-time performance improvement the utility of teleportation-based approaches is unclear.

**Questions:**

1. What does “enhance the gradient norm” mean?
2. In what sense is symmetry teleportation “a state-of-the-art algorithm”?
3. It is strange to cite a 2018 applications paper for MLPs (they have been around for a long time), a 2018 paper proposing ReLUs for classification for ReLUs (which have also been around for a long time), and a 2022 applying Transformers to time series for multi-head attention when they were introduced in 2017 by Vaswani et al.

---

> ### Author Response · Authors · 2024-11-25
>
> **Response to “The approximation scheme is not very clearly described. The goal of the approximation is only defined in words, and the error we should expect is not analyzed”:**
>
> The pseudocode for teleportation with null space gradient projection is already provided in Appendix A.1. Unlike the linear approximation approach proposed by Mishkin et al. (2024), our algorithm does not necessarily involve any form of approximation. When the SVD threshold is set to 1, as in our experiments (except for Penn Treebank, where a threshold of 0.99 is used), the gradient is projected onto the exact null space, ensuring that the loss remains exactly invariant. The SVD threshold serves as a hyperparameter that introduces additional flexibility, and its behavior is empirically analyzed in Section 4.5.
>
> **Response to “A major claim is the improved speed over past teleportation approaches, such as the one using symmetry or the one using linear approximation”:**
>
> Comparison among algorithms in the context of teleportation presents significant challenges. This difficulty arises because the mechanisms for maintaining the invariance of loss during teleportation differ among the approaches, such as the group-action-based method proposed by Zhao et al. (2022), our method, and the approach described in Mishkin et al. (2024). Designing a fair comparison across these baselines is hard. The only reasonable approach we can think of is to compare the methods using their respective best hyperparameter settings. However, grid-searching for the optimal set of hyperparameters is highly impractical, if not infeasible, due to the sensitivity and versatility of teleportation hyperparameters. Moreover, the "best" hyperparameters often involves leveraging significantly more computational resources than a typical or reasonable setting would allow. For instance, Zhao et al. (2022) demonstrated that scheduling five teleportation epochs performs better than one epoch, and using 32 teleportation steps outperforms eight steps. While, in principle, performing as many teleportation steps as possible before each main task update would yield optimal performance, this approach is computationally prohibitive.
>
> Given these considerations, we believe the more critical contribution lies in increasing the applicability of teleportation by designing a more efficient and generalizable algorithm. Accordingly, our experiments focus on comparing the efficiency and generalizability of the methods, which we believe provides a more meaningful and practical evaluation.
>
> **Response to “The observed improvement is mainly in training performance, with minimal improvement in generalization”:**
>
> The primary focus of this paper is to extend the teleportation technique to a broader class of functions while proposing a more efficient algorithm. The use of gradnorm as the teleportation objective function was intended as a demonstration of the effectiveness of our method in accelerating convergence and is not necessarily designed to enhance generalization. We appreciate your observation regarding this point, as it highlights an important branch of research concerning the choice of teleportation loss. As demonstrated in Zhao et al. (2023), the selection of teleportation loss can indeed yield different outcomes; for instance, employing a teleportation loss related to curvature has been shown to improve generalization on test sets.
>
> While our work does not explore teleportation with alternative objective functions that may enhance generalization, we believe that the acceleration in training convergence itself is a noteworthy contribution. As computational efficiency becomes increasingly critical in the era of big data and modern machine learning, our findings address a pressing concern in the field and warrant further attention.
>
> **Response to “What does “enhance the gradient norm” mean?”:**
>
> This simply refers to an "increase" in the gradient norm.
>
> **Response to “In what sense is symmetry teleportation a state-of-the-art algorithm?”:**
>
> To date, there are only two teleportation algorithms: symmetry teleportation (Zhao et al., 2022) and the linear approximation method (Mishkin et al., 2024). None of the existing works compare against any other teleportation algorithms, as conducting a fair comparison is extremely challenging. We consider both methods to be state-of-the-art, as they are the only established algorithms in this domain, and no qualitative performance comparison has been conducted between the two approaches.
>
> **Response to “It is strange to cite a 2018 applications paper for MLPs ”:**
>
> We sincerely thank you for pointing this out. We have carefully reviewed and updated all citations to ensure they are accurate and appropriately representative of the referenced work.

---

### Official Review · Reviewer_z2Na · 2024-11-04

**Soundness:** 2
**Presentation:** 2
**Contribution:** 2
**Rating:** 3
**Confidence:** 4

**Summary:**

The paper presents a teleportation technique for accelerating the convergence of gradient descent. The idea of teleportation is to move the parameters on the level set of the loss function to get a better (e.g., steeper) point before taking a gradient descent step, and this is done every few iterations. The authors aim to address limitations in previous teleportation methods, which were primarily restricted to multi-layer perceptrons (MLPs), by generalizing it to a broader range of architectures, including convolutional neural networks (CNNs) and transformers. Their method aims to reduce computational overhead by eliminating dependency on specific group actions and using an efficient gradient projection technique.

**Strengths:**

- The paper concerns an interesting problem, specifically exploring optimization acceleration through teleportation techniques.
- The introduction effectively provides background context and situates the study within related work, giving readers a clear overview of the existing literature and the motivation for this approach.
- The idea of doing gradient ascent on the gradient norm, while projecting it onto the level set of the original loss is interesting.

**Weaknesses:**

- The authors mix layer-wise and global operations without a clear explanation or justification. While Equations (4) and (5) and Algorithm 1 (correctly) outline parameter updates at a global level, Sections 3.1, 3.2, and 3.3 abruptly pivot to layer-wise operations and projections without justification. The authors should clarify why the projection of the entire parameter vector 𝜋 is equivalent to projecting subsets of the elements (i.e., layers) separately.
- Relatedly, from Equation (5), it appears that 𝜋 is defined to take inputs of the same dimensionality as the complete set of weights. If so, then reusing 𝜋 in Equations (11-13) is incorrect, as these equations seem to imply different dimensionality.
- Again, on a related note, the statement that "the gradient of the teleportation objective function resides within the space spanned by the input data" is incorrect and misleading. The paper actually seems to refer to the input spaces at each layer, not the space spanned by the input data. This claim should be corrected and replaced with a precise statement about the layers if there exists one.

**Questions:**

- Please refer to the weaknesses above.
- In the unnumbered equations between (7) and (8) for MLP, CNN, and Self-Attention: I see that the order of multiplication in which the input to the layer appears differs between architectures (x*\delta in CNN and Self-Attention, as opposed to \delta*x in MLP). Can the authors explain whether this order affects the theoretical claims or the projected outcome, and if so, in what ways?

---

> ### Author Response · Authors · 2024-11-25
> **Response to Official Review of Submission8526 by Reviewer z2Na**
>
> **Response to “The authors mix layer-wise and global operations without a clear explanation or justification”:**
>
> We admit that the shift from the general explanation in Equations (4) and (5) to the layerwise notation in Section 3 and beyond may indeed cause some confusion. It is important to note that the null space gradient projection operation only works at the layerwise level, as each layer's weight is linearly multiplied by the layer's input. Specifically, the gradient of each layer is projected onto the null space of the input of respective layer. This ensures that each layer's output remains invariant, thereby preserving the invariance of the input to the subsequent layer, and ultimately, the final output and the loss of the entire model. To address this, we have revised all relevant expressions to consistently use layerwise notation.
>
> **Response to “Relatedly, reusing 𝜋 in Equations (11-13) is incorrect, as these equations seem to imply different dimensionality”:**
>
> This issue should now be resolved, as we have updated all projections to explicitly use layerwise notation. Additionally, to avoid ambiguity, we have revised the notation for π, adding a subscript l to clearly indicate the layer-specific projection. We have modified the manuscript accordingly.
>
> **Response to “The statement that "the gradient of the teleportation objective function resides within the space spanned by the input data" is incorrect and misleading":**
>
> You are absolutely correct that the statement, along with related ones, all refers to the input data of each specific layer. We initially believed the statement was clear, as we used layerwise notation from Section 3 onward, but we did not fully account for potential confusion arising from the earlier sections. To address this, we have revised the paper to ensure the statements are precise and explicitly refer to each layer where appropriate.
>
> **Response to “Can the authors explain whether this order affects the theoretical claims or the projected outcome, and if so, in what ways?”:**
>
> The order of operations is solely determined by the column definition of x in the MLP and the definition of the matrix X, with row features in CNNs and self-attention, as established in Section 3. If we were to redefine x in the MLP as a row vector, the order of multiplication would be the same as in MLP and self-attention.

---

### Official Review · Reviewer_u2NW · 2024-11-04

**Soundness:** 2
**Presentation:** 3
**Contribution:** 2
**Rating:** 3
**Confidence:** 3

**Summary:**

This paper presents a novel algorithm, "teleportation with null space gradient projection," designed to accelerate optimization in deep learning models. The proposed method improves upon existing teleportation techniques by reducing runtime and mitigating error accumulation. Additionally, it extends teleportation optimization to a broader range of models. The authors validate their approach across diverse model architectures, including Multi-Layer Perceptrons (MLPs), Convolutional Neural Networks (CNNs), and Transformers, employing various benchmark datasets and optimization methods to demonstrate its effectiveness.

**Strengths:**

- The paper is well-written and easy to follow.
- The proposed method is architecturally flexible, as it does not rely on group action from the underlying architecture. This allows the approach to be applicable across a broader range of architectures.

**Weaknesses:**

- The explanation for why projecting onto the Residual Gradient Space (RGS) keeps the parameters within the loss-invariant level set of the original loss is unclear, since the gradient being projected is the gradient of the teleport loss projected to RGS.
- The paper lacks a comparison with other state-of-the-art methods that employ teleportation, such as Zhao et al. (2022) and Mishkin et al. (2024).
- A runtime comparison with non-teleportation optimizers counterparts (SGD, Momentum, Adagrad, Adam) is absent.

**Questions:**

- How does projecting the gradient of the teleportation loss onto RGS ensure the parameters remain on the level set of the original loss?
- Could you provide a performance comparison of the proposed method with other teleportation techniques, specifically comparing with Zhao et al. (2022) on MLP models and with Mishkin et al. (2024) on applicable models?
- What are the runtime of the baseline optimizers before and after introducing teleportation with null-space gradient projection?

---

> ### Author Response · Authors · 2024-11-25
> **Response to Official Review of Submission8526 by Reviewer u2NW**
>
> **Response to “Explanation on null space gradient projection”:**
>
> Updating the model weights with the gradient of the teleportation loss projected onto the RGS effectively ensures that the output of each layer remains invariant. Consequently, the original loss also remains invariant, as it depends solely on the output and the corresponding labels. While the original loss remains unchanged, the teleportation loss can still be minimized during the teleportation step because it is dependent only on the weights, rather than on the outputs of the layers.
>
> **Response to “Lack of comparison with other state-of-the-art methods”:**
>
> To date, there are only two teleportation algorithms: symmetry teleportation (Zhao et al., 2022) and the linear approximation method (Mishkin et al., 2024). None of the existing works compare against any other teleportation algorithms, as conducting a fair comparison is extremely challenging.
>
> This difficulty arises because the mechanisms for maintaining the invariance of loss during teleportation differ among the approaches. Designing a fair comparison across these baselines is hard. The only reasonable approach we can think of is to compare the methods using their respective best hyperparameter settings. However, grid-searching for the optimal set of hyperparameters is highly impractical, if not infeasible, due to the sensitivity and versatility of teleportation hyperparameters. Moreover, the "best" hyperparameters often involves leveraging significantly more computational resources than a typical or reasonable setting would allow. For instance, Zhao et al. (2022) demonstrated that scheduling five teleportation epochs performs better than one epoch, and using 32 teleportation steps outperforms eight steps. While, in principle, performing as many teleportation steps as possible before each main task update would yield optimal performance, this approach is computationally prohibitive.
>
> Given these considerations, we believe the more critical contribution lies in increasing the applicability of teleportation by designing a more efficient and generalizable algorithm. Accordingly, our experiments focus on comparing the efficiency and generalizability of the methods, which we believe provides a more meaningful and practical evaluation.
>
> **Response to “A runtime comparison with non-teleportation optimizers counterparts (SGD, Momentum, Adagrad, Adam) is absent”:**
>
> Since you are specifically referring to non-teleportation optimizers, we assume this pertains to the graphs in Section 4.4, where only the runtime of group-action-based teleportation and our method is included. This is because these graphs focus on the runtime of teleportation steps, which are not applicable to non-teleportation algorithms as they do not involve teleportation runtime.
>
> However, since your comment pertains to different optimizers, it is possible that you are referring to the loss trajectory graphs, which include both teleportation and non-teleportation optimizers.
>
> Alternatively, you may be referencing the wall-clock runtime versus loss graph, which is currently absent for both teleportation and non-teleportation optimizers. Zhao et al. (2022) demonstrated that symmetry teleportation achieves faster convergence in loss versus wall-clock time compared to non-teleportation algorithms, consistent with the trends observed in loss versus epoch graphs. Given that our method has been shown to be significantly more efficient than their symmetry teleportation algorithm, we are confident that our method would also demonstrate superior performance on the loss versus wall-clock time convergence graph.
>
> To address this gap, we will include these graphs across all architectures in the future revised version of the paper.

---

### Official Review · Reviewer_4wRs · 2024-11-09

**Soundness:** 2
**Presentation:** 2
**Contribution:** 3
**Rating:** 6
**Confidence:** 4

**Summary:**

This work considers an alternative strategy for teleportation in optimization landscapes, i.e., going to locations within the level set of the loss but which are better suited for further optimization, like having higher gradient norm. Past methods consider using the group action of continuous symmetries, but this makes them limited to MLPs. This work considers maximizing the gradient norm to teleport while ensuring the layerwise updates of parameters falls in the null space of the layer input. Experiments showcase that this method leads to faster convergence in train and test performance while performing at par with baseline optimization methods.

**Strengths:**

- The idea to utlize (layer) input-space null projection is built atop the work of GPM (Saha et al'21) in continual learning and its use here for the purposes of teleportation is neat.

- This extends the applicability of the approach and might draw further interest in these methods. The resulting method is also more efficient than symmetry teleport based methods.

- The experiments cover a variety of scenarios, wherein the method results in faster convergence while being as good if not slightly better than the baselines.

**Weaknesses:**

- **Wall-clock comparison of convergence:**
 It is unclear how much excess time is being used during the process of teleportation, and if the gains in faster convergence are worth the extra effort.

- **Difference in arrived solutions:**
I would like to see if the solutions reached with teleportation differ qualitatively to those reached without. Can the authors make the LMC curves (Frankle et al, 2019) to see if there are barriers between the reached solutions?

- **Comparison to group action based method:**
 I think it would be interesting to compare the results of your method to group action based ones in a simple setting with MLPs to see in what ways the methods differ.


- **Stability of the hyperparameters**:
It is not clear to me how much do the hyperparameters of teleportation, as well as SVD thresholds, have to be tuned. Can you present a study showing the robustness (or lack of)?

- **Poor referencing of related work:**
A lot of the citations are plain wrong. E.g., I don't think the citations are representative, and some are just plain wrong:

  - Hessian matrix (Sun et al., 2019).
  - Adam (Kashyap 2022)
  - ReLU (Agarap, 2018)
  - MLP (Taud & Mas, 2018).
  - CNN (Li et al., 2021).
  - multi-head self-attention layers (Wen et al., 2022).

**Questions:**

See weaknesses section.

Regarding presentation, I would also make a few suggestions. Instead of input-space, quality it as input space of layers. Otherwise it is confusing. Also, please fix the image titles to TinyImageNet (which currently say ImageNet).

---

> ### Author Response · Authors · 2024-11-25
> **Response to Official Review of Submission8526 by Reviewer 4wRs**
>
> **Response to “Wall-clock comparison of convergence”:**
>
> Zhao et al. (2022) demonstrated that the convergence of symmetry teleportation with respect to loss versus wall-clock time is faster compared to non-teleportation algorithms, consistent with the trends observed in the loss versus epoch graph. Since our results have shown that our method significantly outperforms their symmetry teleportation algorithm in terms of efficiency, it can be reasonably inferred that our approach would exhibit superior performance on the loss versus wall-clock time convergence graph as well. We acknowledge the importance of presenting these graphs and will incorporate them in appendix in the future revised version of the paper to provide a comprehensive comparison.
>
> **Response to “Difference in arrived solutions”:**
>
> The primary focus of this paper is to extend the teleportation technique to a broader class of functions while proposing a more efficient algorithm. The use of gradnorm as the teleportation objective function was intended as a demonstration of the effectiveness of our method in accelerating convergence and is not necessarily designed to enhance generalization. We appreciate your observation regarding this point, as it highlights an important branch of research concerning the choice of teleportation loss. As demonstrated in Zhao et al. (2023), the selection of teleportation loss can indeed yield different outcomes; for instance, employing a teleportation loss related to curvature has been shown to improve generalization on test sets.
>
> While our work does not explore teleportation with alternative objective functions that may enhance generalization, we believe that the acceleration in training convergence itself is a noteworthy contribution. As computational efficiency becomes increasingly critical in the era of big data and modern machine learning, our findings address a pressing concern in the field and warrant further attention.
>
> **Response to “Stability of the hyperparameters”:**
>
> The sensitivity of teleportation hyperparameters is indeed high, as noted in the discussion section. For instance, Zhao et al. (2022) demonstrated that performance can improve significantly by scheduling more teleportation epochs and steps. The hyperparameters used in our experiments primarily follow the conventions established in their work. The sensitivity of hyperparameters also highlights the need to carefully study the tradeoff between performance and computational cost, which makes algorithmic efficiency even more critical. Therefore, the primary motivation behind our work is to provide an algorithm that is significantly more efficient than existing approaches. Nevertheless, we will include additional ablation studies in the future revised version of the paper.
>
> The ablation studies on the threshold of SVD is present in section 4.5.
>
> **Response to “Comparison to group action based method”:**
>
> To date, there are only two teleportation algorithms: symmetry teleportation (Zhao et al., 2022) and the linear approximation method (Mishkin et al., 2024). None of the existing works compare against any other teleportation algorithms, as conducting a fair comparison is extremely challenging.
>
> This difficulty arises because the mechanisms for maintaining the invariance of loss during teleportation differ among the approaches. The only reasonable approach we can think of is to compare the methods using their respective best hyperparameter settings. However, grid-searching for the optimal set of hyperparameters is highly impractical, if not infeasible, due to the sensitivity and versatility of teleportation hyperparameters. Moreover, the "best" hyperparameters often involves leveraging significantly more computational resources than a typical or reasonable setting would allow. For instance, Zhao et al. (2022) demonstrated that scheduling five teleportation epochs performs better than one epoch, and using 32 teleportation steps outperforms eight steps. While, in principle, performing as many teleportation steps as possible before each main task update would yield optimal performance, this approach is computationally prohibitive.
>
> Given these considerations, we believe the more critical contribution lies in increasing the applicability of teleportation by designing a more efficient and generalizable algorithm. Accordingly, our experimental section focuses on comparing the efficiency and generalizability of the methods, which we believe provides a more meaningful and practical evaluation.
>
> **Response to “Poor referencing of related work”:**
>
> We sincerely thank you for pointing this out. We have carefully reviewed and updated all citations to ensure they are accurate and appropriately representative of the referenced work.
>
> Additionally, we have revised the input space to be explicitly described as the layerwise input space for improved clarity. We have also corrected the title to reflect "Tiny ImageNet" instead of "ImageNet" for consistency.

---

### Meta-Review · Area_Chair_s6m2 · 2024-12-19

**Metareview:**

This paper proposes a new optimization algorithm for neural networks. The approach is based on the idea of "teleportation" which combines optimizing the loss function with searching around locations with equal loss to find one potentially more amenable to optimization. The paper generalizes the teleportation method beyond MLPs to CNNs and Transformers and can achieve faster convergence while matching final performance. Experiments across multiple architectures show the method performs well and some theory backs up these results.

The reviewers identified issues with the work including; lacking experimental details, weak baselines, the paper's clarity and incorrect/missing citations.

While reviewers enjoyed parts of the work, I believe their issues/concerns with outweigh and I will recommend rejection.

**Additional Comments On Reviewer Discussion:**

Initially reviewers had negative views of the work citing weaknesses written above and asked authors to clarify parts of the work they found confusing. Authors responded to their concerns, added explanation, and added additional experiments. Despite this, reviewers' opinion of the work was not notably changed.

---

### Decision · Program_Chairs · 2025-01-22

Reject